



# Aerosol emission factors from traditional biomass cookstoves in India: Insights from field measurements

Apoorva Pandey[1], Sameer Patel[1], Shamsh Pervez[2], Suresh Tiwari[3], Gautam Yadama[4,a], Judith C. Chow[5], John G. Watson[5], Pratim Biswas[1], Rajan K. Chakrabarty[1]

[1]Department of Energy, Environmental and Chemical Engineering, Washington University in St. Louis, St. Louis, MO 63130, USA
[2]School of Studies in Chemistry, Pandit Ravishankar Shukla University, Raipur, Chhattisgarh 492010, India
[3]Indian Institute of Tropical Meteorology, Pune, Maharashtra 411008, India
[4]Brown School of Social Work, Washington University in St. Louis, St. Louis, MO 63130, USA
[5]Divison of Atmospheric Sciences, Desert Research Institute, Reno, NV 89512, USA
[a] now at: School of Social Work, Boston College, Boston, MA 02467, USA

*Correspondence to*: Rajan K. Chakrabarty (chakrabarty@wustl.edu) and Apoorva Pandey (apoorva@wustl.edu)

**Abstract.** Residential solid biomass cookstoves are important sources of aerosol emissions in India. Cookstove emission rates
are largely based on laboratory experiments conducted using the standard water-boiling test, but real-world emissions are often higher owing to different stove designs, fuels, and cooking methods. Constraining mass emission factors (EFs) for prevalent cookstoves is important because they serve as inputs to bottom-up emission inventories used to evaluate health and climate impacts. Real-world EFs were measured during winter, 2015, for a traditional cookstove ("*chulha*") burning fuel-wood (FW), agricultural residue (AG) and dung (DG) from different regions of India. Average ($\pm$ 95% confidence interval) EFs for FW,
AG, and DG were: 1) PM$_{2.5}$ mass: 6.8 (4.7 – 9.4) g kg$^{-1}$, 7.1 (3.9 – 11.8) g kg$^{-1}$, and 14.5 (7.5 – 25.3) g kg$^{-1}$, respectively; 2) elemental carbon (EC): 0.6 (0.4 – 0.9) g kg$^{-1}$, 1.0 (0.4 – 2.0) g kg$^{-1}$, and 0.6 (0.3 – 1.3) g kg$^{-1}$, respectively; and 3) Organic carbon (OC): 3.1 (2.0 – 4.6) g kg$^{-1}$, 4.5 (2.3 – 8.0) g kg$^{-1}$, and 8.2 (4.2 – 15.01) g kg$^{-1}$, respectively. The mean ($\pm$ 95% confidence interval) OC-to-EC mass ratios were 6.5 (4.5 – 9.1), 7.6 (4.4 – 12.2), and 12.7 (8.8 – 17.8), respectively, with OC and EC quantified by the IMPROVE_A thermal/optical reflectance protocol. These real-world EFs are higher than those from
laboratory-based measurements. Combustion conditions have larger effects on EFs than the fuel-types. We also report the carbon mass fractions of our aerosol samples determined using the thermal-optical reflectance method. The mass fraction profiles are consistent between the three fuel categories, but markedly different from those reported in past literature.

## 1 Introduction

The Indian subcontinent is a regional hotspot for anthropogenic emissions (Ramanathan and Carmichael, 2008).
Carbonaceous aerosol (black carbon – BC, and organic carbon – OC) in India is linked to surface dimming (Kambezidis et al., 2012), solar warming of the lower atmosphere (Ramanathan et al., 2001;2007), changing regional monsoon patterns (Chung and Seinfeld, 2005;Menon et al., 2002;Ramanathan et al., 2005), and accelerated melting of Himalayan glaciers (Ramanathan



et al., 2007). Particulate matter (PM) emissions—particularly particles with aerodynamic diameters less than 2.5 µm ($PM_{2.5}$)—are also associated with numerous adverse consequences for human health (Pope et al., 2009;Pope and Dockery, 2006). The

Global Burden of Disease study has identified indoor air pollution as the largest risk factor and outdoor air pollution as the seventh largest risk factor for disability-adjusted life years in India (Murray et al., 2013) .

The most recent emissions inventory for India indicated that residential biomass cookstoves are the largest contributors to total annual $PM_{2.5}$ emissions (Pandey et al., 2014;Sadavarte and Venkataraman, 2014). In 2010, 67% of Indian households, more than 160 million total, relied primarily on solid fuels for cooking (Census, 2011). Commonly used cooking

appliances are mostly mud stoves, with some three-stone type brick stoves and metal stoves (Kar et al., 2012), that burn fuel-wood (FW), agricultural residues (AG), and dried cattle dung (DG). Traditional cookstoves have low combustion efficiencies, resulting in incomplete combustion and high PM emissions (Smith et al., 2000b).

Emissions performance of cookstoves is commonly expressed in terms of mass-basis emission factors (EFs) or mass of pollutant emitted per unit mass of fuel burned. PM emission rates depend on fuel properties, combustion device, operator

behaviour and cooking patterns (Leavey et al., 2015;Sahu et al., 2011;Roden et al., 2009). Cookstove heating efficiencies and EFs and are often measured in a laboratory setting using a water-boiling test (WBT) with high- (boiling) and low- (simmering) power phases (Habib et al., 2008;Smith et al., 2000a;MacCarty et al., 2008). These standardized tests are useful for comparing different stove-fuel combinations, but they do not represent real-world stove behaviors found in the field (Roden et al., 2006;Roden et al., 2009;Smith, 2007). Habib et al. (2008) changed the amount of water boiled from 0.5 kg to 1.5 kg in the

WBT test, thereby changing the fuel burn rate and burn cycle duration, and observed a factor of ~2.7 increase in the $PM_{2.5}$ EF and a factor of ~2 increase in the OC fraction. A real-world study of Honduran wood-burning cookstoves (Roden et al., 2006) found higher $PM_{2.5}$ EFs and OC content than those from previous laboratory studies. Roden et al. (2006;2009) found that real-world fire tending and cooking practices (and therefore burn conditions) were important factors determining PM EFs and compositions.

Real-world EFs for commonly used fuel types and cooking technologies in India are needed for accurate of bottom-up emission estimates (Bond et al., 2013;Bond et al., 2004;Pandey et al., 2014). Inventoried emission rates serve as inputs to regional and global atmospheric transport models that predict spatiotemporal profiles of pollutant burdens and associated impacts on climate and human health (Bond et al., 2013;Guttikunda and Calori, 2013;Sadavarte et al., 2016;Schulz et al., 2006). Alternatively, these inventories are used in conjunction with impact metrics such as intake fraction (Grieshop et al.,

2011) and global warming potential (Shindell et al., 2012) to evaluate mitigation policies (MHFW, 2015;Sagar et al., 2016). Such measurements identify key parameters to be monitored during laboratory testing and appliance certification.

With the above goals, we measured cookstove emission characteristics in a rural Indian household. Local meals were prepared with a traditional mud stove ("*chulha*") using biomass fuels collected from different regions of India. Real-time measurements of emitted gas concentrations were conducted and PM filter samples were collected at regular time intervals

during each cooking cycle. $PM_{2.5}$, OC and EC EFs are reported here as a function of fuel-type and combustion phase. Thermal carbon fractions provided by the IMPROVE_A protocol are also examined.



## 2 Methods

Thirty separate cooking tests were conducted between December 19 and 30th of 2015 in a rural household on the outskirts of Raipur, a city located in the central Indian state of Chhattisgarh (abbreviated as Chh.). ~77% of Chhattisgarh
households are located in rural areas and rely almost entirely on solid biomass fuels for cooking (Census, 2011). On a national level, fuel-wood, agricultural residue and dung are used as primary cooking fuels by 49%, 9% and 8% Indian households respectively (Census, 2011). Accounting for average combustion efficiencies and calorific values of these fuels, annual fuel usage estimates are 250 MT fuel-wood, 73 MT agricultural residue and 100 MT dung (Pandey et al., 2014). For this study, fuel-wood was obtained from Uttar Pradesh (U.P.), Rajasthan (Raj.), Andhra Pradesh (A.P.), Bihar, and Punjab which
collectively account for 35% of the total fuel-wood user base in India. All wood fuels were in the form of chunks with typical dimensions of 5 – 15 cm. Cattle dung (in the form of dung cakes dried in the sun) was collected from U.P. and Bihar, which use 60% of India's total for cooking. Agricultural residues from of *tur* crops (a type of woody stalk) and rice straw were procured from a village near the study location. Test fuels were collected and stored in sealed bags, and later analysed for elemental (carbon, oxygen, hydrogen, nitrogen) composition and moisture content Fuel compositions are compared in Table
1. Per real-world practice, fuel samples were naturally dried in the sun and stored indoors, bringing moisture contents to <9%. These compositions are consistent with those reported in other tests (Habib et al., 2008;Smith et al., 2000a).

**Table 1: Elemental composition and moisture content of the biomass fuels in this study.**

| Fuel | Elemental composition (%) | | | | Moisture content (%) |
|:---:|:---:|:---:|:---:|:---:|:---:|
| | Carbon | Hydrogen | Oxygen | Nitrogen | |
| U.P. dung | 33.1 | 4.0 | 30.0 | 1.6 | 7.5 |
| Bihar dung | 41.4 | 5.1 | 33.6 | 2.1 | 8.6 |
| Chh. rice straw | 40.7 | 5.5 | 39.0 | 0.8 | 5.3 |
| Chh. *tur* stalk | 48.4 | 6.5 | 42.7 | 0.6 | 4.8 |
| Punjab wood | 50.3 | 0.2 | 40.9 | 0.4 | 6.2 |
| Raj. Wood | 49.7 | 5.6 | 42.9 | 0.1 | 8.1 |
| U.P. wood | 49.9 | 0.1 | 41.8 | 0.2 | 5.6 |





| | | | | | |
|---|---|---|---|---|---|
| A.P. wood | 48.3 | 0.1 | 43.4 | 0.7 | 3.1 |

Table 2 describes the fuels used and the foods cooked; replicate tests were made for some of these combinations with
at least three for each fuel. Dung (20-50 g) was doused with approximately 10 ml kerosene for initial ignition and the test fuel
was added after a steady flame was achieved. Additional fuel of the same type was added as needed to complete the recipe. A
ten-minute period following lighting of the fire or the addition of kindling materials is designated as *ignition* phase. The
remainder of the cooking cycle was designated as the *flaming* phase when a visible flame was present. Combustion entered
the *smoldering* phase when the flame died down. The U.P. dung and Chh. rice straw could not sustain the flaming phase for
more than a few minutes. Dung is typically smoldered for low-power cooking applications, and it is used as kindling material
for igniting fuel-wood in a typical rural household. The low carbon content of U.P. dung (Table 1) possibly hinders its ability
to sustain a flame, more so than Bihar dung. Rice straw has a low material density and high surface-to-volume ratio, and
therefore tends to burn out very quickly. It also produces large amounts of smoke, making its use as a standalone fuel
impractical and harmful for the cook's health. To circumvent these limitations, a few experiments established a steady flame
using U.P. dung/Chh. rice straw mixed with U.P. wood (approximately 2.5:1 ratio of test fuel mass to wood mass).





**Table 2: List of cooking experiments conducted during the 10-day intensive study period. Abbreviations for Indian states: U.P. = Uttar Pradesh, Raj. = Rajasthan, A.P. = Andhra Pradesh, Chh. = Chhattisgarh.**

| Day | Primary fuel used | No. of replicate experiments | Food cooked |
|---|---|---|---|
| 1 | Bihar dung | 1 | lentil-rice |
| 2 | U.P wood | 2 | rice, vegetables |
| 3 | U.P wood | 1 | tea |
| | Raj. wood | 4 | lentils, rice, vegetables |
| 4 | U.P. wood | 2 | lentils |
| | A.P. wood | 1 | rice |
| 5 | A.P. wood | 2 | rice, vegetables |
| | Bihar dung | 1 | tea |
| 6 | Chh. *tur* stalks | 3 | rice, vegetables, tea |
| 7 | U.P. dung[a] | 3 | vegetables, rice |
| | Bihar dung | 1 | tea |
| 8 | Chh. rice straw[a] | 3 | rice, vegetables |
| | Chh. wood | 1 | tea |
| 9 | Raj. wood | 4 | water heated, rice and curry |
| 10 | Punjab wood | 1 | milk porridge |

[a] Two experiments for each fuel conducted with fuel-wood mixed with the test fuel

The emission testing system is shown in Fig. 1. An eight-armed stainless steel probe (based on Roden et al. (2006)) sampled naturally-diluted emissions at ~1.2 m above the top of the stove. Each arm of probe was 0.5 m in length, with 4 uniformly placed holes facing the plume.  This probe was connected to three real-time instruments−a Kanomax Portable

Mobility Particle Sizer (PAMS) (Kulkarni et al., 2016), a TSI Sidepak (Zhu et al., 2007), and a Testo-350 gas analyser (Wang et al., 2012). The PAMS recorded particle size distributions from 10 – 400 nm mobility diameter.  The Sidepak provides a light-scattering (670 nm) surrogate for measured $PM_{2.5}$ mass that is calibrated with Arizona Road Dust (O'Shaughnessy and Slagley, 2002).  PM concentrations exceeded the top range of these instruments for short periods during the plume monitoring. The Testo-350 gas analyser was factory-calibrated prior to the experiments for carbon monoxide (CO) and carbon dioxide

($CO_2$).  Measured concentrations (acquired every second) were at least five times the detection limits of 1 ppm CO and 0.01 % $CO_2$ by volume.  $PM_{2.5}$ was collected on 47 mm Teflon-membrane and pre-baked quartz-fiber filters several times during a cooking cycle using Minivol (5 L min$^{-1}$) samplers (AirMetrics Model 4.2) with greased impactor inlets located in the plume





~0.9m above the stove. Filter sample durations ranged from 0.5 to 4 minutes, based on the continuous SidePak reports, to prevent filter overloading. Field blanks were collected (minimum sampling duration of 15 minutes) each day before testing.

The Teflon filters were weighed before and after sampling to obtain the net mass deposit which was divided by the sample volume (flow rate times duration) to obtain the concentration. Quartz filters were analysed using the Interagency Monitoring of Protected Visual Environments – A (IMPROVE_A) thermal-optical reflectance (TOR) method (Chow et al., 2007b;2011) to determine elemental and organic carbon fractions in the sampled particulates. .

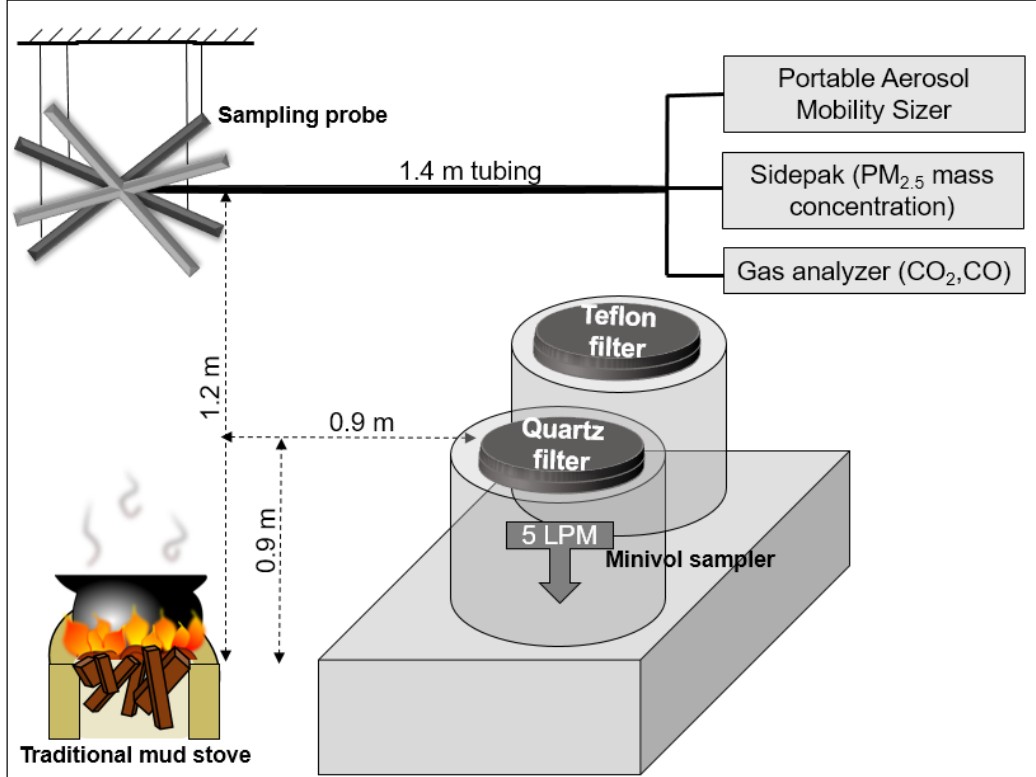

**Figure 1: Schematic representation of the experimental setup.**

Using the carbon mass balance technique, fuel-based EFs were calculated for each filter:

$$EF_i = CMF_{fuel} \frac{c_i}{\Delta C_{CO_2}\left(\frac{M_C}{M_{CO_2}}\right) + \Delta C_{CO}\left(\frac{M_C}{M_{CO}}\right)} \qquad (1)$$

where $EF_i$ is the EF of species $i$ in grams emitted per gram of fuel consumed. $CMF_{fuel}$ is the carbon mass fraction of the fuel, which ranged from 33% to 50% for the tested fuels. $C_i$ is the concentration of emittant $i$, in this case $PM_{2.5}$, OC, or EC, in g

$m^{-3}$, determined for each Teflon and quartz filter. $\Delta C_{CO2}$ and $\Delta C_{CO}$ are the concentrations above ambient levels of $CO_2$ and CO in g $m^{-3}$, respectively. $M_C$, $M_{CO2}$, and $M_{CO}$ are the atomic or molecular weights of C, $CO_2$, and CO in g $mole^{-1}$. Eq. (1) assumes that the carbon emitted in $CH_4$, NMHC, and PM is negligible compared to that in CO and $CO_2$.





Wireless optical particle sensors (details available in Patel et al. (2017)) were attached to the Minivol sampler and the sampling probe during four experiments to check for any significant differences in the particle concentrations measured at the two locations. A correction factor of 1.04 was applied based on the correlation between the sensor measurements at the two locations (Supplemental Data). The equation above was also corrected to account for the small fraction of fuel carbon that gets converted to gaseous volatile organic carbon, rather than $CO_2$ or CO, assumed as 2.4% (Habib et al., 2008;Roden et al., 2006).


## 3 Results and discussion

Figure 2 compares EFs for the different fuels. There are no statistically significant (unpaired *t*-test, 95% confidence intervals) EF differences for wood-fuels from different regions of India. Bihar dung EFs exceeded those for U.P. dung, possibly owing to the addition of wood to sustain flaming. On average, $PM_{2.5}$ and OC emission factors for dung were higher than those for fuel-wood. EFs for dung, rice straw and *tur* stalk show a larger spread EFs for fuel-wood.







**Figure 2: Box plots of (a) PM$_{2.5}$ emission factors, (b) OC emission factors, and (c) EC emission factors. All units are g-pollutant kg$^{-1}$-fuel. Boxes denote lower and upper quartiles; whiskers are 1.5 times the interquartile ranges of the upper and lower quartiles. The numbers in paranthesis in panel (c) indicate the number of samples for each fuel.**



Figure 3 compares EFs for the different burning phases. $PM_{2.5}$ EFs are highest during the ignition phase for all fuels. The OC/EC ratio (Fig. 3b) increases from ignition, to flaming, to smoldering for all fuels, with the highest ratios

found during smoldering. CO EFs and modified combustion efficiencies (MCE, ratio of $CO_2$ concentration to $CO+CO_2$ concentration) show no correlation with $PM_{2.5}$ emission factors or OC-to-EC ratios (Supplemental Data). MCEs exceeded 0.9 – a value associated with flaming (Reid et al., 2005;Zhang et al., 2008) – for nearly 90% of the test durations, even when no flaming was observed.

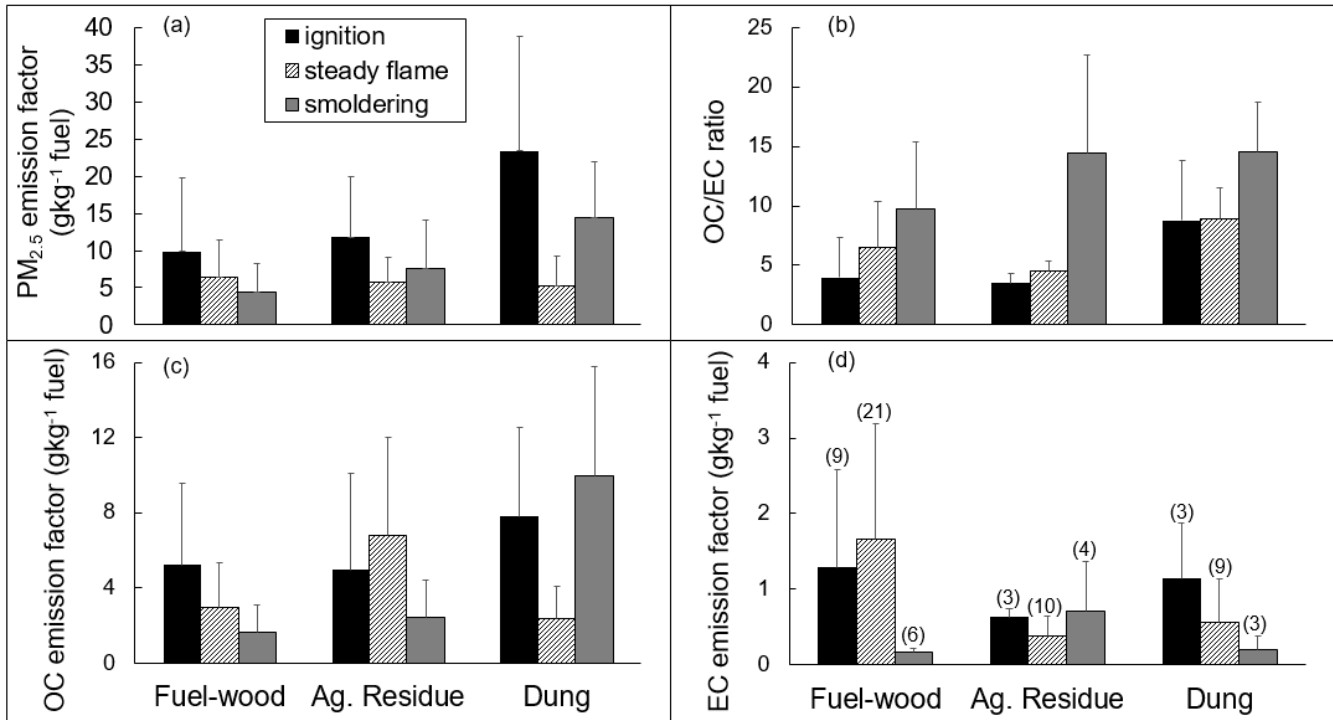

**Figure 3: Fuel-wise average values of (a) $PM_{2.5}$ emission factors, (b) OC-to-EC ratios, (c) OC emission factors, and (d) EC emission factors, categorized by observed combustion phases. One-sided error bars are shown to denote one standard deviation from the mean. The numbers in paranthesis in panel (d) indicate the number of samples for each fuel and combustion phase.**

Average EFs for the entire burn cycle were calculated as a time-weighted sum of EFs for each phase of combustion. Fuel-wood and agricultural residue are used predominantly in flaming conditions to carry out the bulk of cooking operations,

resulting in weights of 17% ignition, 66% steady flame and 17% smoldering. For dung, the weights are 17% ignition, 17% flaming and 66% smoldering because dung is used for longer low-power operations such as heating water/milk and roasting vegetables. These EFs are compared with other reported EFs in Figure 4. Average fuel-wood $PM_{2.5}$ EFs and OC/EC ratios in this study are comparable to those reported by (Roden et al., 2006) for Honduran cooking, but they are about twice as large as those reported for laboratory studies (Habib et al., 2008;MacCarty et al., 2008;Saud et al., 2012). For agricultural residue

and dung, the average EFs and OC/EC ratios are 1.1 – 3.8 times and 1.3 – 3.6 times higher, respectively, compared to those reported by Saud et al. (2012) and Habib et al. (2008).





**Figure 4: Average PM$_{2.5}$ emission factors and OC-to-EC ratios for the three fuel categories in this study, compared with relevant studies. Error bars for values estimated in this study denote 95% confidence intervals based on standard errors of the means. Error**
**bars for other studies are the bounds reported within those studies.**

Thermal fractions of total carbon constituted by the IMPROVE_A protocol are compared in Fig. 5. OC1, OC2, OC3 and OC4 refer to carbon that evolves at temperatures of 120 ⁰C, 250 ⁰C, 450 ⁰C, and 550 ⁰C respectively, in the inert helium atmosphere. OP denotes pyrolyzed carbon, OC charring in the inert helium carrier. EC1, EC2 and EC3 fractions evolve in a 2%O$_2$/98%He oxidizing atmosphere at 550 ⁰C, 700 ⁰C and 800 ⁰C, respectively. Fig.5 compares fractions from this study
with those reported for controlled biomass (hardwood and softwood) burning reported in Chow et al. (2007a) and fuel-wood cookstove emissions from Indian laboratory tests CPCB (2011). OC3 was the most abundant fraction, ~50% of OC3 by mass, while the profiles in literature ranged 10%-34% in the OC3 fraction. The OC1 fraction for all fuels in this study was uniformly less than 3%, a finding comparable only to the 5% OC1 reported for softwood, but not for the other two profiles.





Carbonaceous aerosol source profiles have been used for source apportionment, and they may also have implications for climate and health impact assessments. In an previous study (Pandey et al., 2016), we reported that light absorbing OC may play a larger role in light absorption by cookstove emissions than that from earlier work on biomass burning in the U.S.A. The difference in constituents of OC emissions from the two sources might contribute to the observed difference in their optical characteristics, since thermal stability is known to be inversely related to the light absorption efficiency of organic compounds (Andreae and Gelencsér, 2006;Saleh et al., 2013).

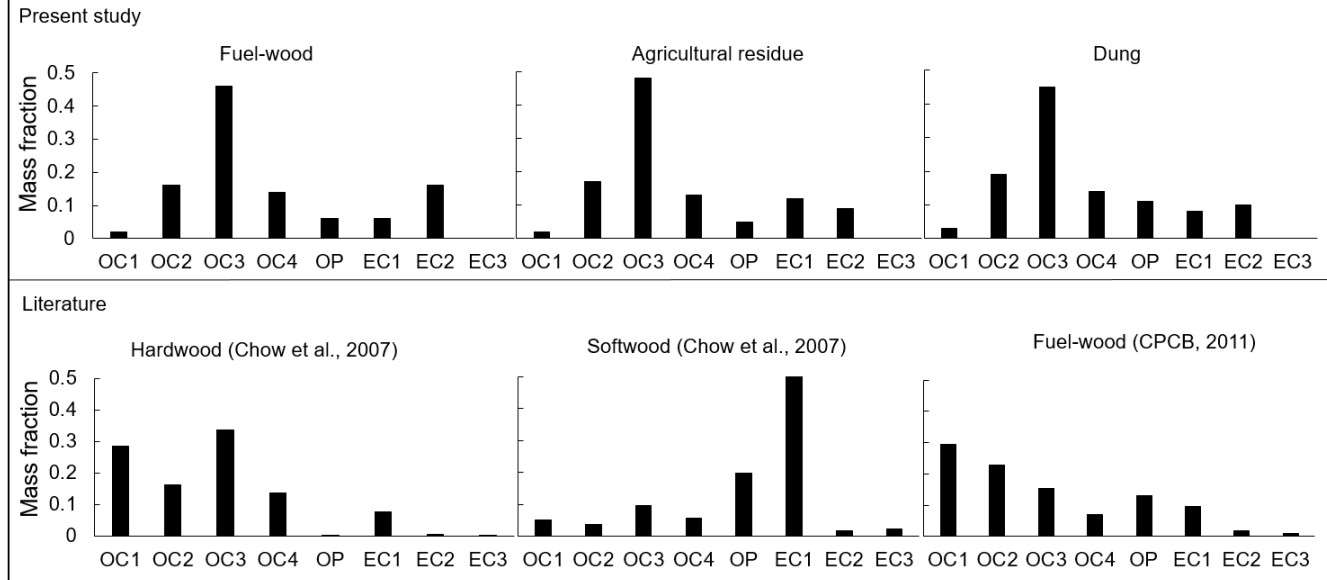

**Figure 5: Fraction of total carbon contributed by the IMPROVE_A thermal carbon fractions.**

## 4 Conclusions

We estimated PM$_{2.5}$, OC and EC mass emission factors from real-world combustion of commonly-used biomass fuels in India. Fuel-wood and dung EFs are more than twice those derived for laboratory tests for similar appliances and fuels. Wood fuels yielded similar EFs, despite their being gathered from widely separated parts of India.   A short period of time immediately following the ignition phase showed the highest and most variable PM$_{2.5}$ EFs for all fuels. The highest OC-to-EC ratios were observed during smoldering (no visible flame present). The OC3 thermal fraction contributed more than 50% of the total carbon, indicating that most of the PM emissions are non-evaporative.  The thermal fractions found it this study differ from those found in other biomass burning tests.  Our findings suggest that combustion conditions may have a larger influence on intrinsic properties of biomass combustion emissions than fuel variability. Connecting aerosol emissions from a given source to their effects on climate and human health requires knowledge of their mass emission rates and physiochemical





properties. We believe the EFs from this study would contribute toward improving evaluations of climate and health impacts of carbonaceous aerosols over India.


*Acknowledgements*

This work was partially supported by the National Science Foundation under Grant No. AGS1455215, NASA ROSES under Grant No. NNX15AI66G, and the International Center for Advanced Renewable Energy and Sustainability (I-CARES) at Washington University in St. Louis. The authors thank Madhuri Verma, Rakesh Sahu, and Jeevan Matawale

from Pandit Ravishankar Shukla University, and Praveen Kumar from Washington University in St Louis, for their help with fuel and sample collection.

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
