# Peer review of "Aerosol emission factors from traditional biomass cookstoves in India: Insights from field measurements"

_Atmospheric Chemistry and Physics, 2017_

## Referee Comment (RC1) · Anonymous Referee #2 · 21 Jun 2017

In the manuscript "Aerosol emission factors from traditional biomass cookstoves in India: Insights from field measurements" the real-world emission factors of PM components from different biomass fuels in food cooking are studied in an Indian household. The findings increase the understanding of real-world PM, OC, and EC emission factors for a wide range of combustion of solid fuels from different regions. The results are compared with previous findings in standardized tests, and in general, the results of the manuscript show higher PM emissions and OC/EC ratios existing in the literature. In a big picture, the main findings are novel and significant enough for an ACP paper but the quality of presentation needs improvement (major revision). On the positive side, I like the title, abstract and conclusions parts of the manuscript.

[Figure]

Specific comments

Table 2 lists the experiments performed, indicating the number of reps and the foods that were cooked. The variation of the foods cooked was a bit random; apparently, the family was just living ordinary life. In the analysis, the food is not considered as a factor instead it is just causing deviation to different EFs for each fuels. Authors should analyze the role of the food cooked while discussing the results. Is the food itself emitting something, or is it only so that different foods require different amount of heat or time?

Authors have used several instruments in the tests: PAMS, Sidepak analyzer, gas analyzer, minivol sampler and wireless optical particle sensors. First of all, locations of wireless optical sensors are not shown in Figure 1. Also it is not clear why minivol sampler could not use the "eight armed sampling probe" and adjacent tubing to be in parallel with other instruments. Figure 1 would be more informative if the whole room or house would be also sketched to have an idea how the exhaust flows in the space. I assume there is not a dedicated stack for exhaust gas ventilation.

The authors have corrected values from different sampling locations using wireless sensors and correction factor of about 1.04 (data shown in Figure S1). About half of the time both sensors were showing maximum concentrations, so that time they were not showing any meaningful results. Was the maximum concentrations also included in the regression analysis? I see that at many times the difference between sensors was much more than 4%. Also you should include all the experiments with wireless sensors to this regression analysis.I think getting this factor right is very crucial thing because gas analyzers measured from different location than the minivol sampler. It would be also interesting to see how gas concentrations change during combustion/non-combustion phases to see how much is the increase in CO and CO2 concentrations in the measurement locations. Authors could include a plot of this in the supplementary information.

The manuscript in current form presents only data from filter collections (Minivol sampler) while the authors also list other instruments that were used. For instance, authors should show time series of some individual test how the PM emission behaves during different phases of the combustion. Overall the role of combustion phase is highlighted in the results. Authors should also show and analyze particle size distributions measured with the PAMS instruments. This would help to e.g. identify whether the PM mass is in the particle size range of PAMS or above it.

The results are focusing on the effects of combustion phase and fuel quality of the EFs. The discussion and conclusions of the results are so far in form where authors just focus on the measured values. The cause and effect analysis is much lacking in the current version of the manuscript. For instance, why different phases of combustion show different results? Why are the deviations in results such high? The authors should dedicate more effort in this.

Minor remarks

Figure quality. You should remove all borderlines from the figures. Font quality is bad in Figures 2 and 4.

Row 79. Full stop apparently missing.

Rows 131-133. You basically don't have to use value of 2.4% here since you have measured OC/EC yourself, or you can also compare the amount of non-CO or non-$CO_2$ carbon based on your results to this value.

Row 135. "Figure 2 compares EFs for the different fuels." Are these EF's from the "whole burning event" including all the phases from combustion.

Row 156. "is used". Should this be "was used"?

Figure 5. I imagine you could have all the bars in the same plot.

Row 185. "Fuel-wood and dung EFs. . ." You could rephrase this to be in more accurate

form, it is not emission of fuel-wood or dung after all.

Row 188. OC3 does not look to be more than 50% in Fig. 5.
* * *

---

## Referee Comment (RC2) · Anonymous Referee #1 · 7 Jul 2017

This manuscript presents emission factors for three types of solid fuels from India based on real-world tests and associated chemical speciation analysis. The results are novel, and provide relevant information on EFs related with indoor combustion in the Indian (and the broader South Asian) context. The manuscript is written well, and provides succinct information on the sampling methods and previous work on the topic. The observations are described well, but the results are not discussed in detail.

Please include a brief description of the sampling location in the methods section- dimensions of the room where sampling was conducted, ventilation type (or lack thereof) in the sampling area, and representativeness of the sampling location.

[Figure]

Lines 115-118: Please include limits of detection for carbon analysis using IM-PROVE_A protocol, and average particle mass based on gravimetric analysis.

Each emission test included data on PM (size distribution, real-time PM data with Side-pak, sensors), and gases (CO, CO2) in addition to filter samples, but none of the data is discussed in the manuscript. Please consider including the profile of real-time emissions for a sample test for each fuel. Also, include average particle size, or representative particle size distributions for the different regimes (ignition, flaming and smoldering).

With respect to the differences between EFs for Bihar and UP dung, is it known if there are differences in composition of dung cakes?

Figure 5: It is interesting to note that the carbon profile for the present study is very different from the results in the CPCB (2011) report. It would be beneficial if the authors could comment on the reason for the differences (methods, sample type or something else?). Also, what could be causing OC3 fraction to contribute nearly 50% of the total OC mass? Please elaborate.

Based on the observations from this analysis, what are the implications for the real-world? Are we able to derive substantial conclusions that could be relevant for mitigation policies, or exposure reduction measures?

Lines 154-157: Does this mean that one household could use different fuel types depending on the type of cooking activity being undertaken?

Figure S1: Using a plot of PM concentrations from the optical sensors will be much more useful compared to the raw signal data currently included in the supplemental data. Was a secondary check performed to compare average concentration of Sharp sensor for each cooking event with gravimetric filter measurement or Sidepak measurements?

Minor comments:

[Figure]

Figure 2: Please change the resolution. The image gets blurred if the reader zooms in (same problem in case of Figure 4).

Revise the manuscript - words and/or punctuation marks are missing at some places (e.g., Lines 77, 79 and 138).

In the supplement, the sentence "MCE is typically treated as an identifier of . . . . . . " seems to be incomplete. Please edit.

Line 89: The thermal fractions "found in" rather than "found it"
* * *

---

## Author Comment (AC1) · 25 Aug 2017

This manuscript presents emission factors for three types of solid fuels from India based on real-world tests and associated chemical speciation analysis. The results are novel, and provide relevant information on EFs related with indoor combustion in the Indian (and the broader South Asian) context. The manuscript is written well, and provides succinct information on the sampling methods and previous work on the topic. The observations are described well, but the results are not discussed in detail.

[Figure]

Comment: Please include a brief description of the sampling location in the methods section- dimensions of the room where sampling was conducted, ventilation type (or lack thereof) in the sampling area, and representativeness of the sampling location.

Response: A schematic of the room has been included in the Supplemental Data and a brief description was added to the main text.

Revision at Line114: The test kitchen (Fig. S1) was on the first floor of house, separate from all other rooms. A permanently open door was the only entry from an open terrace. A partially covered window covered was a second ventilation source.

Comment: Lines 115-118: Please include limits of detection for carbon analysis using IMPROVE_A protocol, and average particle mass based on gravimetric analysis.

Response: The following details were added to the main text.

Revision at Line 132: The mass of Teflon filter deposits ranged 50 – 300 $\mu$g.

Revision at Line 134: The minimum detection limits of the TOR analysis are about 9 $\mu$g for OC and 1 $\mu$g for EC (Solomon et al., 2014).

Comment: Each emission test included data on PM (size distribution, real-time PM data with Sidepak, sensors), and gases (CO, CO2) in addition to filter samples, but none of the data is discussed in the manuscript. Please consider including the profile of real-time emissions for a sample test for each fuel. Also, include average particle size, or representative particle size distributions for the different regimes (ignition, flaming and smoldering).

Response: A sample plot of real-time gas concentration and Sidepak particle mass concentrations is now shown in the Supplemental Data.

Comment: With respect to the differences between EFs for Bihar and UP dung, is it known if there are differences in composition of dung cakes?

Response: Bihar dung had a higher carbon content than U.P. dung (Table 1), any other

differences in composition are not known.

Comment: Figure 5: It is interesting to note that the carbon profile for the present study is very different from the results in the CPCB (2011) report. It would be beneficial if the authors could comment on the reason for the differences (methods, sample type or something else?). Also, what could be causing OC3 fraction to contribute nearly 50% of the total OC mass? Please elaborate.

Response: The Central Pollution Control Board (CPCB) of India conducted source apportionment studies in six Indian cities (CPCB, 2011). As part of this study, a source profile was developed for PM2.5 emissions from wood chulhas, through laboratory combustion tests. Further details of the burn cycle or sampling system used in this study are not known, but the source profile obtained has been uploaded to their website: http://www.cpcb.nic.in/Source_Apportionment_Studies.php. Although we cannot comment on the reason for the differences, we presented the findings because, to our knowledge, this is the only available profile of Indian cookstove emissions and has been used in the apportionment studies mentioned above.

Revision at Line 188: We also compare our results with a source profile developed for PM2.5 emissions from wood chulhas, as part of source apportionment studies conducted by the Central Pollution Control Board (CPCB) of India (CPCB (2011)). This profile was based on laboratory burns, but details of the test fuel and burn protocol are not known.

Comment: Based on the observations from this analysis, what are the implications for the real world? Are we able to derive substantial conclusions that could be relevant for mitigation policies, or exposure reduction measures?

Response: The conclusions section has been expanded with more analysis of the effect of fuel type and combustion phase on emission rates. Further discussion of the implications of our findings has also been added. We cannot draw any conclusions as to exposure reduction or mitigation.

Revision at Line 206: Earlier studies have shown that fuel burn rate (Habib et al., 2008) and fire-tending practices (Roden et al., 2009) affect cookstove emissions, therefore standardized burn protocols (typically a water boiling test) cannot replicate cookstove performance in the field. In a controlled test, variability in wood size could be expected to affect emissions from fuel-wood combustion, hence, laboratory studies typically used identical wood chunks. The four wood fuel types in this study were used as chunks in a range of sizes, and yielded similar EFs even though these samples were gathered from widely separated parts of India. Overall, the variability in emissions between multiple measurements for the same fuel was larger than or comparable to the differences in average emission rates for different fuels.

Revision at Line 213: A short period immediately following the ignition phase showed the highest and most variable PM2.5 EFs for all fuels, consistent with previous observations of high emissions from lighting and refueling of stoves (Roden et al., 2009). The use of variable amounts of kerosene and other kindling materials, as required in any burn, may explain the large spread in emission measurements during ignition. The highest OC-to-EC ratios were observed during smoldering (no visible flame present), when pyrolysis is the dominant process.

Revision at Line 217: The OC3 thermal fraction contributed nearly 50% of the total carbon mass for emissions from all fuels, indicating that most of the PM emissions are non-evaporative. This could have implications for the climatic impact of cookstove emissions, especially since volatility of OC emissions is inversely related to light absorption efficiency in the near-UV and visible wavelengths (Andreae and Gelencsér, 2006). The thermal fractions found in this study differ from those in other biomass burning tests, including the source profile used an input to receptor models of particulate emissions in India (CPCB, 2011).

Revision at Line 225: The EFs from this study could contribute toward improving evaluations of climate and health impacts of carbonaceous aerosols over India. Further investigation is needed on the relationship between the composition of OC emissions

and their effects on climate and health.

Comment: Lines 154-157: Does this mean that one household could use different fuel types depending on the type of cooking activity being undertaken?

Response: Yes, it is anecdotally known that a single household often uses a mix of fuels (Venkataraman et al., 2010). The household in this study also had an LPG stove but they continued to use freely available biomass (dung and wood) to reduce their cooking costs. However, available large-scale survey data (Census, 2011) only presents the distribution of primary cooking fuels across households.

Comment: Figure S1: Using a plot of PM concentrations from the optical sensors will be much more useful compared to the raw signal data currently included in the supplemental data. Was a secondary check performed to compare average concentration of Sharp sensor for each cooking event with gravimetric filter measurement or Sidepak measurements?

Response: A separate objective of this study was to validate the use of wireless sensors in the field (Patel et al., 2017). Reference measurements by the Sidepak were well-correlated with data from a collocated sensor. However, the Sidepak measurement being a measure of particle light scattering does not reflect the actual particle concentration, unless the instrument is independently calibrated for the given type of particles. Therefore, in this manuscript, we have not converted the sensor signals to equivalent Sidepak concentrations.

Minor comments: Figure 2: Please change the resolution. The image gets blurred if the reader zooms in (same problem in case of Figure 4). Revise the manuscript - words and/or punctuation marks are missing at some places (e.g., Lines 77, 79 and 138). In the supplement, the sentence "MCE is typically treated as an identifier of . . .. . . " seems to be incomplete. Please edit. Line 89: The thermal fractions "found in" rather than "found it"

Response: These comments have been addressed.

Anonymous Referee #2

In the manuscript "Aerosol emission factors from traditional biomass cookstoves in India: Insights from field measurements" the real-world emission factors of PM components from different biomass fuels in food cooking are studied in an Indian household. The findings increase the understanding of real-world PM, OC, and EC emission factors for a wide range of combustion of solid fuels from different regions. The results are compared with previous findings in standardized tests, and in general, the results of the manuscript show higher PM emissions and OC/EC ratios existing in the literature. In a big picture, the main findings are novel and significant enough for an ACP paper but the quality of presentation needs improvement (major revision). On the positive side, I like the title, abstract and conclusions parts of the manuscript.

Specific comments Comment: Table 2 lists the experiments performed, indicating the number of reps and the foods that were cooked. The variation of the foods cooked was a bit random; apparently, the family was just living ordinary life. In the analysis, the food is not considered as a factor instead it is just causing deviation to different EFs for each fuel. Authors should analyze the role of the food cooked while discussing the results. Is the food itself emitting something, or is it only so that different foods require different amount of heat or time?

Response: See and Balasubramanian (2008) investigated emissions during various cooking operations on a gas stove stove. On average, they found particle mass concentrations of about 23 $\mu$g/m3 from boiling and about 65 $\mu$g/m3 from deep frying, showing that certain cooking processes could emit a significant number of particles. Particle emissions from gas stoves are low enough that the effect of the cooking process itself can be observed. Solid biomass cookstoves are known to emit over 5-10 times more particles than gas stoves under identical lab settings (Smith et al., 2000). In this study, we estimated particle mass concentrations of 500 $\mu$g/m3-20 mg/m3, from gravimetric measurements. Therefore, the contribution from food emissions is expected to be negligible.

Comment: Authors have used several instruments in the tests: PAMS, Sidepak analyzer, gas analyzer, minivol sampler and wireless optical particle sensors. First of all, locations of wireless optical sensors are not shown in Figure 1. Also it is not clear why minivol sampler could not use the "eight armed sampling probe" and adjacent tubing to be in parallel with other instruments. Figure 1 would be more informative if the whole room or house would be also sketched to have an idea how the exhaust flows in the space. I assume there is not a dedicated stack for exhaust gas ventilation.

Response: The Minivol samplers being ambient PM2.5 sensors could not be connected to the tubing from the sampling probe. Given the constraints of the small kitchen, we tried to ensure that all instruments excepting the Minivol samplers be sampled from the same location (that is via the eight armed probe). The position of the wireless sensors is now shown in Figure 1. A schematic of the room has been included in the Supplemental Data, and described briefly in the manuscript.

Revision at Line 114: The test kitchen (Fig. S1) was on the first floor of house, separate from all other rooms. A permanently open door was the only entry from an open terrace. A partially covered window covered was a second ventilation source.

Comment: The authors have corrected values from different sampling locations using wireless sensors and correction factor of about 1.04 (data shown in Figure S1). About half of the time both sensors were showing maximum concentrations, so that time they were not showing any meaningful results. Was the maximum concentrations also included in the regression analysis? I see that at many times the difference between sensors was much more than 4%. Also you should include all the experiments with wireless sensors to this regression analysis. I think getting this factor right is very crucial thing because gas analyzers measured from different location than the minivol sampler.

Response: We used the optical wireless sensors for a total of 6 runs. Following the reviewer's comments, the regression analysis was re-evaluated: we observed that data points close to the saturation limit of the sensors were strongly influencing the regression results. The following changes were made

Revision in Supplemental Data: The saturation voltage for the sensors is close to 750 mV, discarding all values higher than 750 mV, regression analysis of the remaining points yields a slope of 0.96. However, if the saturation threshold was set at 745 mV, the slope changed to 0.89. This is probably because saturation behavior for these sensors is a soft-limit saturation, such that the input-response relationship becomes non-linear at some voltage lower than the final limiting value of 750 mV. If measurements from this non-linear region are included, the linear regression analysis would give erroneous results. Therefore, we systematically reduced the threshold values until we observed negligible change in the regression slope. Finally, we discarded the data points where either of the sensors had readings above the linearity threshold (720 mV). About 60% of all data points were used, and a slope of 0.63 (R2=0.65) was obtained. The scatterplot of data from the two sensors is shown in the Supplemental Data.

Revision at Line 145: Wireless optical particle sensors (details available in Patel et al. (2017)) were attached to the Minivol sampler and the sampling probe during six experiments to check for any significant differences in the particle concentrations measured at the two locations. Measurements where either sensor was saturated were discarded, and a linear regression analysis performed on the valid data points. A correction factor of 1.6 was applied based on regression slope (Supplemental Data).

Comment: It would be also interesting to see how gas concentrations change during combustion/non-combustion phases to see how much is the increase in CO and CO2 concentrations in the measurement locations. Authors could include a plot of this in the supplementary information. Response: A sample plot of real time gas concentration and Sidepak particle mass concentrations is also now shown in the Supplemental Data.

Comment: The manuscript in current form presents only data from filter collections (Minivol sampler) while the authors also list other instruments that were used. For instance, authors should show time series of some individual test how the PM emission behaves during different phases of the combustion. Overall the role of combustion phase is highlighted in the results. Authors should also show and analyze particle size distributions measured with the PAMS instruments. This would help to e.g. identify whether the PM mass is in the particle size range of PAMS or above it.

Response: We were unable to draw any conclusions from the PAMS data because the instrument was typically saturated within the first ten minutes of a burn.

Comment: The results are focusing on the effects of combustion phase and fuel quality of the EFs. The discussion and conclusions of the results are so far in form where authors just focus on the measured values. The cause and effect analysis is much lacking in the current version of the manuscript. For instance, why different phases of combustion show different results? Why are the deviations in results such high? The authors should dedicate more effort in this.

Response: The conclusions section has been expanded with more analysis of the effects of fuel type and combustion phase on emission rates. Further discussion on the implications of our findings has also been added.

Revision at Line 206: Earlier studies have shown that fuel burn rate (Habib et al., 2008) and fire-tending practices (Roden et al., 2009) affect cookstove emissions, therefore standardized burn protocols (typically a water boiling test) cannot replicate cookstove performance in the field. In a controlled test, variability in wood size could be expected to affect emissions from fuel-wood combustion, hence, laboratory studies typically used identical wood chunks. The four wood fuel types in this study were used as chunks in a range of sizes, and yielded similar EFs even though these samples were gathered from widely separated parts of India. Overall, the variability in emissions between multiple measurements for the same fuel was larger than or comparable to the differences in

average emission rates for different fuels.

Revision at Line 213: A short period immediately following the ignition phase showed the highest and most variable PM2.5 EFs for all fuels, consistent with previous observations of high emissions from lighting and refueling of stoves (Roden et al., 2009). The use of variable amounts of kerosene and other kindling materials, as required in any burn, may explain the large spread in emission measurements during ignition. The highest OC-to-EC ratios were observed during smoldering (no visible flame present), when pyrolysis is the dominant process.

Minor remarks Figure quality. You should remove all borderlines from the figures. Font quality is bad in Figures 2 and 4.

Response: Higher quality figures are now embedded in the manuscript.

Row 79. Full stop apparently missing.

Response: Corrected.

Rows 131-133. You basically don't have to use value of 2.4% here since you have measured OC/EC yourself, or you can also compare the amount of non-CO or nonCO2 carbon based on your results to this value.

Response: We have measured OC/EC but we cannot estimate non-(CO and CO2) carbon because we do not know the mass of fuel burned as a function of time. We cannot balance the fuel carbon with carbonaceous emissions.

Row 135. "Figure 2 compares EFs for the different fuels." Are these EF's from the "whole burning event" including all the phases from combustion.

Response: Yes, this figure includes all data points for each fuel.

Row 156. "is used". Should this be "was used"? Figure 5. I imagine you could have all the bars in the same plot. Row 185. "Fuel-wood and dung EFs. . ." You could rephrase this to be in more accurate, it is not emission of fuel-wood or dung after all. Row 188.

[Figure]

OC3 does not look to be more than 50% in Fig. 5

Response: The four comments above have been addressed.

References

Census: Houselisting and Housing Census Data New Delhi., 2011.

CPCB: Air quality monitoring, emission inventory and source apportionment study for Indian cities. National Summary Report, Central Pollution Control Board, 2011.

Patel, S., Li, J., Pandey, A., Pervez, S., Chakrabarty, R. K., and Biswas, P.: Spatio-temporal measurement of indoor particulate matter concentrations using a wireless network of low-cost sensors in households using solid fuels, Environ. Res., 152, 59-65, 2017.

See, S. W., and Balasubramanian, R.: Chemical characteristics of fine particles emitted from different gas cooking methods, Atmospheric Environment, 42, 8852-8862, 2008.

Smith, K. R., Uma, R., Kishore, V., Lata, K., Joshi, V., Zhang, J., Rasmussen, R., Khalil, M., and Thorneloe, S.: Greenhouse gases from small-scale combustion devices in developing countries, Phase IIa: Household Stoves in India, US Environmental Protection Agency, Research Triangle Park, NC, 98, 2000.

Venkataraman, C., Sagar, A., Habib, G., Lam, N., and Smith, K.: The Indian national initiative for advanced biomass cookstoves: the benefits of clean combustion, Energy for Sustainable Development, 14, 63-72, 2010.

Please also note the supplement to this comment:
https://www.atmos-chem-phys-discuss.net/acp-2017-291/acp-2017-291-AC1-supplement.pdf

—————————————————

[Figure]

**Supplement:**

Supplement of

**Aerosol emission factors from traditional biomass cookstoves in India: Insights from field measurements**

Apoorva Pandey et al.

*Correspondence to*: Rajan K. Chakrabarty (chakrabarty@wustl.edu) and Apoorva Pandey (apoorva@wustl.edu)

**Kitchen layout**

The test kitchen was located on second floor of a house. It had one window located on the front wall, a U-shaped mud stove was fixed to this wall. The only door to the room was left fully open during all burns. All real-time instruments were placed in the back of the room, their inlets connected, via conductive tubing, to an eight-armed steel probe. Two Minivol samplers collected $PM_{2.5}$ samples on Teflon and quartz filters.

[Figure]

**Figure S1: Schematic layout (top-view) of the kitchen**

**Wireless sensor data**

Measurements from Sharp GP2Y sensors attached to the sampling probe (Sensor 1) and to the Minivol sampler (Sensor 2), from day 9 of the study, are shown in Figure S1. These sensors include an infrared emitting diode, the emission from which is scattered by the particles, and a phototransistor converts the scattered light to a voltage output proportional to the PM concentration.

[Figure]

**Figure S2: Raw signals from the PM sensors located at the sampling probe (Sensor1) and the Minivol PM$_{2.5}$ sampler (Sensor2).**

The saturation voltage for the sensors is close to 750 mV, discarding all values higher than 750 mV, regression analysis of the remaining points yields a slope of 0.96. However, if the saturation threshold was set at 745 mV, the slope changed to 0.89. This is probably because saturation behavior for these sensors is a soft-limit saturation, such that the input-response relationship becomes non-linear at some voltage lower than the final limiting value of 750 mV. If measurements from this non-linear region are included, the linear regression analysis would give erroneous results. Therefore, we systematically reduced the threshold values until we observed negligible change in the regression slope. Finally, we discarded the data points where either of the sensors had readings above the linearity threshold (720 mV). About 60% of all data points were used, and a slope of 0.63 (R$^2$=0.65) was obtained. Therefore, the concentration measured by the Minivol sampler was adjusted upwards by a factor of 1.6 (=1/0.63).

[Figure]

**Figure S3: Scatter plot of measurements from Sensor1 and Sensor 2. Linear regression provided a slope of 0.63, with an R$^2$ of 0.65.**

**Real-time measurements**

A sample plot of real-time particle and gas concentration profiles from day 9 of the study. Please note that the Sidepak instrument does not measure actual particle mass concentration, but instead measures light scattering at 670 nm wavelength and provides an equivalent concentration of Arizona Test Dust that would produce the same magnitude of light scattering.

Over a period of two hours, Sidepak PM measurements and CO concentration (solid in panel B) fluctauated every few minutes. Sidepak was saturated at an equivalent concentration of 20 µg/m3, giving the appearance of a steady state. Re-fueling typically caused a sudden change in particulate and CO emissions.

[Figure]

**Figure S4: Real-time measurements of (A) Sidepak PM$_{2.5}$ mass concentrations in µg/m$^3$, and (B) CO concentrations (solid) in and CO$_2$ concentration (dashed), both in ppm.**

**Carbon Monoxide (CO) emission factors**

Emission factors of CO were calculated using the equation below:

$$EF_{CO} = CMF_{fuel} \frac{C_{CO}}{\Delta C_{CO_2}\left(\frac{M_C}{M_{CO_2}}\right) + \Delta C_{CO}\left(\frac{M_C}{M_{CO}}\right)}$$

where $EF_{CO}$ is the CO emission factor (g of CO released per kg of fuel burnt), $CMF_{fuel}$ is the carbon mass fraction of the fuel, which ranged from 33% to 50% for the tested fuels. $C_{CO}$ is the concentration of CO in g m$^{-3}$. $\Delta C_{CO2}$ and $\Delta C_{CO}$ are the concentrations above ambient levels of $CO_2$ and CO in g m$^{-3}$, respectively. $M_C$, $M_{CO2}$, and $M_{CO}$ are the atomic or molecular weights of C, $CO_2$, and CO in g mole$^{-1}$.

[Figure]

**Figure S5: Fuel-wise average values of CO emission factors, categorized by observed combustion phases. One-sided error bars are shown to denote one standard deviation from the mean.**

Both CO and PM$_{2.5}$ are products of incomplete combustion are their mass emission rates measured during lab cookstove tests are found to correlate (Roden et al., 2009). In this study, no correlation was observed between the estimated CO emission factors and corresponding PM$_{2.5}$ emission factors. Further, we plotted modified combustion efficiencies (MCE), calculated as the ratio of $CO_2$ concentration to CO+$CO_2$ concentration, against OC-to-EC ratios. MCE is typically treated as an identifier of combustion phase, with values greater than 0.9 associated with (Reid et al., 2005; Zhang et al., 2008). We found estimated MCE values above 0.9 for roughly 90% of all run time, even when no flaming phase was visibly observed. They showed no correlation with OC-to-EC ratios.

[Figure]

**Figure S6: Comparisons of (a) CO vs PM$_{2.5}$, EFs and (b) OC/EC ratios vs modified combustion efficiency (MCE) values.**

**References**

Reid, J., Koppmann, R., Eck, T. and Eleuterio, D. (2005). A review of biomass burning emissions part II: intensive physical properties of biomass burning particles. *Atmospheric Chemistry and Physics,* 5, 799-825.
Roden, C. A., Bond, T. C., Conway, S., Pinel, A. B. O., MacCarty, N. and Still, D. (2009). Laboratory and field investigations of particulate and carbon monoxide emissions from traditional and improved cookstoves. *Atmospheric Environment,* 43, 1170-1181.
Zhang, H., Ye, X., Cheng, T., Chen, J., Yang, X., Wang, L. and Zhang, R. (2008). A laboratory study of agricultural crop residue combustion in China: emission factors and emission inventory. *Atmospheric Environment,* 42, 8432-8441.